# Shifting meiotic to mitotic spindle assembly in oocytes disrupts chromosome alignment

Isma Bennabi[1], Isabelle Quéguiner[1], Agnieszka Kolano[2], Thomas Boudier[3] (iD), Philippe Mailly[1], Marie-Hélène Verlhac[1,†,*] (iD) & Marie-Emilie Terret[1,†,**] (iD)

## Abstract

Mitotic spindles assemble from two centrosomes, which are major microtubule-organizing centers (MTOCs) that contain centrioles. Meiotic spindles in oocytes, however, lack centrioles. In mouse oocytes, spindle microtubules are nucleated from multiple acentriolar MTOCs that are sorted and clustered prior to completion of spindle assembly in an "inside-out" mechanism, ending with establishment of the poles. We used HSET (kinesin-14) as a tool to shift meiotic spindle assembly toward a mitotic "outside-in" mode and analyzed the consequences on the fidelity of the division. We show that HSET levels must be tightly gated in meiosis I and that even slight overexpression of HSET forces spindle morphogenesis to become more mitotic-like: rapid spindle bipolarization and pole assembly coupled with focused poles. The unusual length of meiosis I is not sufficient to correct these early spindle morphogenesis defects, resulting in severe chromosome alignment abnormalities. Thus, the unique "inside-out" mechanism of meiotic spindle assembly is essential to prevent chromosomal misalignment and production of aneuploidy gametes.

**Keywords** chromosome misalignment; HSET; meiosis; mitosis; spindle morphogenesis
**Subject Category** Cell Cycle

## Introduction

Animal cells generally assemble mitotic spindles using an "outside-in" mechanism that relies on centrosomes acting as dominant microtubule-nucleating centers (MTOCs). The two centrosomes define the spindle poles and thus the spindle axis along which chromosome segregation will take place at anaphase [1,2]. Oocytes however lack canonical centrosomes, the centrioles being lost before the meiotic divisions occur [3,4]. Interestingly, it was shown in *Drosophila* that maintaining functional supernumerary centrioles during female meiotic divisions leads to abnormal meiosis and aborted embryonic development [5], highlighting the fact that centriole loss is essential for successful sexual reproduction. In mouse oocytes, microtubules are nucleated from chromatin and multiple acentriolar microtubule-organizing centers (aMTOCs) composed of pericentriolar material [6–9]. These aMTOCs are perinuclear before meiotic divisions and fragment at NEBD (nuclear envelope breakdown) to become evenly distributed around chromatin [10,11]. Following NEBD, microtubules become nucleated and stabilized first around chromatin, forming a microtubule ball, and then organized into a stable central array via microtubule motors and microtubule-associated proteins, which sort and orient the microtubules [12–17]. aMTOCs are then progressively sorted along this central array [16]. Following spindle bipolarity setup, the aMTOCs become clustered to establish the spindle poles [17]. Meiotic spindles in oocytes are thus assembled "inside-out".

Spindle assembly in oocytes is a very slow process. Spindle bipolarization is achieved by 4 h in mice [12,13] and by around 7 h in humans [18], thus occupying about half the transition time from NEBD to anaphase in these species. It mirrors the long duration of the first meiotic division, as meiosis I requires 8–12 h in mice and more than 20 h in humans [18]. In addition, whereas spindle poles are organized by two centrosomes in mitosis, pole formation is different in meiosis. In mouse oocytes, poles are organized by multiple aMTOCs. Thus, meiotic spindle poles are often less focused than mitotic ones, having this typical barrel-shaped aspect. Are these unique "inside-out" spindle assembly and organization required for meiotic spindle function, that is, segregating chromosomes? To answer this question, we switched meiotic spindle assembly toward a more mitotic-like mode, with rapid bipolarity and focused pole assembly, and looked at chromosome alignment and segregation. To do so, deregulation of HSET levels was used as a tool to alter early stages of spindle morphogenesis. The kinesin-14 HSET is a minus-end-directed microtubule cross-linking motor important for regulating spindle assembly, spindle length, and pole organization [19–25]. During

1 Center for Interdisciplinary Research in Biology (CIRB), College de France, CNRS, INSERM, PSL Research University, Equipe labellisée FRM, Paris, France
2 International Institute of Molecular and Cell Biology, Warsaw, Poland
3 Université Pierre et Marie Curie, Sorbonne Universités, Paris, France
  *Corresponding author. Tel: +33144271082; E-mail: marie-helene.verlhac@college-de-france.fr
  **Corresponding author. Tel: +33144271692; E-mail: marie-emilie.terret@college-de-france.fr
  †These authors contributed equally to this work as senior authors

mitosis, HSET can slide anti-parallel microtubules apart and sort them into parallel bundles [26–28]. In contrast, when the orientation of two opposing microtubules is parallel, HSET cross-links them and transports them to the poles [23,26]. We show here that a slight increase in HSET levels accelerates spindle formation, in particular spindle bipolarization and aMTOCs clustering.

Importantly, this leads to severe chromosome alignment abnormalities. In an unexpected manner, the unusual length of meiosis I (8 h) is not sufficient to correct early spindle morphogenesis defects, contributing to chromosome misalignment and mis-segregation. Thus, the unique "inside-out" spindle assembly and organization prevent aneuploidy in female gametes.

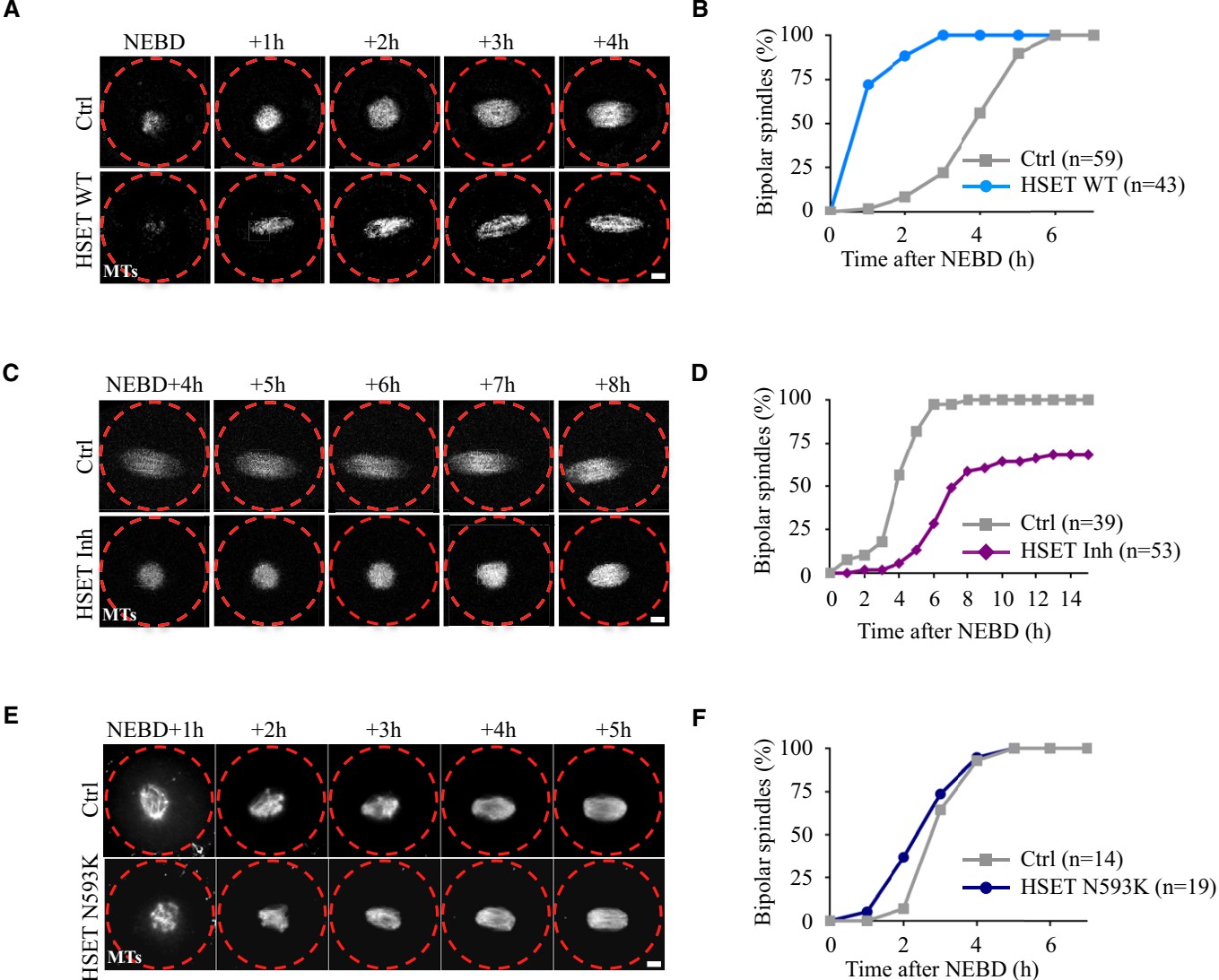

**Figure 1. Modification of the timing of spindle bipolarization.**

A Time-lapse spinning disk confocal microscopy of oocytes expressing GFP-EB3 (gray) alone (Ctrl, upper panel) or together with HSET WT (lower panel). Spindle bipolarization is advanced in HSET WT expressing oocytes compared to controls. Scale bar 10 μm.

B Graph showing the kinetics of spindle bipolarization in controls (gray squares) vs. HSET WT oocytes (blue dots). The kinetics of bipolarization is accelerated in oocytes overexpressing HSET WT compared to controls.

C Time-lapse spinning disk confocal microscopy of oocytes expressing GFP-EB3 (gray) treated (HSET Inh, lower panel) or not (Ctrl, upper panel) with the HSET inhibitor AZ82. Spindle bipolarization is delayed in oocytes inhibited for HSET. Scale bar 10 μm.

D Graph showing the kinetics of spindle bipolarization in controls (gray squares) vs. oocytes inhibited for HSET (purple diamonds). The kinetics of bipolarization is delayed in oocytes inhibited for HSET compared to controls.

E Time-lapse spinning disk confocal microscopy of control oocytes (Ctrl, upper panel) and oocytes expressing HSET N593K (HSET N593K, lower panel). Spindle bipolarization is slightly advanced in oocytes overexpressing HSET N593K compared to controls. All oocytes were incubated with SiR-Tubulin (gray). Scale bar 10 μm.

F Graph showing the kinetics of spindle bipolarization in controls (gray squares) vs. HSET N593K oocytes (dark blue dots). The kinetics of bipolarization is modestly affected in oocytes overexpressing HSET N593K compared to controls.

# Results

### Altering the timing of spindle bipolarization

To modify spindle morphogenesis, we developed an HSET gain-of-function approach. The localization of endogenous HSET was first analyzed in mouse oocytes, by performing immunofluorescence experiments on fixed oocytes. We found that endogenous HSET is localized on the spindle in meiosis I (Fig EV1A, left panel). HSET dynamics and localization were followed in living oocytes, by expressing an exogenous GFP-tagged HSET wild-type (WT) construct. Our exogenous GFP-HSET WT probe displayed the same spindle localization as endogenous HSET (Fig EV1A, middle panel, immunofluorescence) and remained associated with the spindle throughout meiosis I (Fig EV1B, live microscopy). However, HSET WT exogenous expression must be tightly controlled, since too much of it induced spindle collapse and mono-aster formation (see Materials and Methods). We therefore performed experiments with a maximum HSET WT overexpression of 1.6-fold in the whole oocyte (Fig EV1C, immunofluorescence quantification) corresponding to a 4.2-fold accumulation of HSET in the spindle (Fig EV1D, immunofluorescence quantification). Meiotic spindle assembly in the context of an HSET WT overexpression was analyzed by time-lapse spinning disk microscopy. In controls, microtubules formed bipolar spindles within ~4 h after NEBD (Fig 1A, upper panel). In contrast, spindle bipolarization took place much more rapidly in oocytes overexpressing HSET WT (Fig 1A, lower panel and B), skipping the microtubule ball stage described in Ref. [13]. Indeed, the average time of bipolarization setup was achieved in 4 h and 3 min in controls compared to 1 h and 19 min in oocytes overexpressing HSET WT (Figs 1B and EV2A).

The gain-of-function analysis was complemented with an HSET loss-of-function approach. To do so, oocytes were treated with AZ82, a small molecule inhibitor of HSET [29,30], and meiotic spindle assembly was followed using time-lapse spinning disk microscopy. Spindle bipolarization was delayed in HSET-inhibited oocytes (HSET Inh) compared to controls (Ctrl, Fig 1C), requiring 6 h and 55 min in these oocytes (Figs 1D and EV2A). The delay of spindle bipolarization observed with AZ82 could be phenocopied using another allosteric inhibitor of HSET, CW069 (Fig EV2C and

D), structurally unrelated to AZ82 [31]. Taken together, and in contrast to previously published observations [20], these results suggest that HSET levels modulate the timing of meiotic spindle bipolarity in meiosis I.

To understand how HSET drastically impacts the timing of spindle bipolarization, we took advantage of a GFP-HSET mutant N593K (HSET N593K) that can cross-link but does not slide microtubules [23]. GFP-HSET N593K localized on the spindle (Fig EV1A, right panel, immunofluorescence) and had similar distribution along the spindle as GFP-HSET WT (compare Fig EV1B and EV1E, live microscopy). It reached even higher expression levels at NEBD+7h (Fig EV1F, quantification of live microscopy). The timing of spindle bipolarization was only slightly advanced in oocytes overexpressing HSET N593K compared to controls (Figs 1E and F, and EV2B). This suggests that, for the most part, changes in the timing of spindle bipolarization require microtubule sliding by HSET.

### Accelerating spindle pole assembly

Because spindle bipolarization occurs precociously in oocytes overexpressing HSET WT, we next analyzed the consequences of its overexpression on sorting of the aMTOCs. This process occurs concomitant with spindle bipolarization and is followed by aMTOC clustering, which allows spindle pole focusing. To do this, the behavior of aMTOCs was followed by time-lapse microscopy, using mCherry-Plk4 (Polo-like kinase 4) as a marker [32]. We performed an automated 3D analysis of aMTOCs within the spindle. For that, we developed a Fiji plug-in that converts images obtained using live microcopy to binary images and in 3D finds the spindle poles and calculates the distance of each aMTOC to the closest pole (see Materials and Methods; Fig EV3A). In addition, this plug-in allows extraction of the number and distribution of the aMTOCs together with spindle measurements (length, central width, spindle pole width). The measurements were performed at three time points during meiosis I, spanning the critical steps of spindle morphogenesis in controls (Figs 2A and EV3C, middle panels).

At NEBD+1h30, microtubules form a ball, with aMTOCs dispersed around it [13]. At NEBD+4h30, spindle bipolarization is achieved and a robust central array of microtubules allows the progressive sorting of aMTOCs to the poles [16]. At NEBD+6h30,

---

**Figure 2.  Acceleration of aMTOCs sorting and clustering.**

A  Spinning disk confocal microscopy images showing spindle region magnifications of oocytes overexpressing HSET WT, controls and oocytes inhibited for HSET at NEBD+1h30, +4h30, and +6h30. All oocytes express GFP-EB3 (green) and mCherry-Plk4 (red). Scale bar 10 μm.

B  aMTOC sorting in oocytes overexpressing HSET WT (blue dots) and controls (gray dots) at NEBD+1h30, +4h30, and +6h30. The dot plot represents the standard deviation of the repartition of aMTOCs along the axis of the spindle for each oocyte analyzed. Each dot represents an oocyte; the number of oocytes analyzed is written in parentheses. Statistical significance of differences is assessed with a *t*-test with Welch correction where needed: *$P$-value = 0.018, **$P$-value = 0.002. As shown on the scheme, when aMTOCs are not sorted, the standard deviation is high; in contrast, when aMTOCs are sorted to the poles, the standard deviation is low.

C  aMTOCs clustering in oocytes overexpressing HSET WT (blue dots) and controls (gray dots) at NEBD+1h30, +4h30, and +6h30. The dot plot represents the number of aMTOCs per oocyte. Each dot represents an oocyte; the number of oocytes analyzed for each condition is written in parentheses. Statistical significance of differences is assessed with a *t*-test with Welch correction: *$P$-value = 0.011, **$P$-value = 0.003, ***$P$-value < 0.0001.

D  Super resolution images of aMTOCs using SIM, in fixed controls and HSET WT expressing oocytes (pericentrin antibody: gray). Scale bar 5 and 2 μm.

E  Quantification of aMTOCs volume from SIM super-resolution images. Control oocytes gray dots and HSET WT expressing oocytes blue dots. Statistical significance of differences is assessed with a *t*-test with Welch correction: *$P$-value = 0.0453.

F  FRAP analysis of SiR-Tubulin in controls (gray) and in oocytes overexpressing HSET WT (blue) at NEBD+6h30. SiR-Tubulin was photobleached at spindle poles, and its fluorescence recovery was followed. The SiR-Tubulin fluorescence intensity was normalized so that 1 corresponds to the prebleached value and 0 corresponds to the value at the first time point after bleaching. For a single exponential recovery model, the halftime to fluorescence recovery in controls oocytes is $t_{1/2}$ = 62 s compared to $t_{1/2}$ = 55 s for oocytes overexpressing HSET WT. Data are represented as mean ± SD. Statistical significance of differences for the $t_{1/2}$ is assessed with a Mann–Whitney test: $P$-value = 0.87.

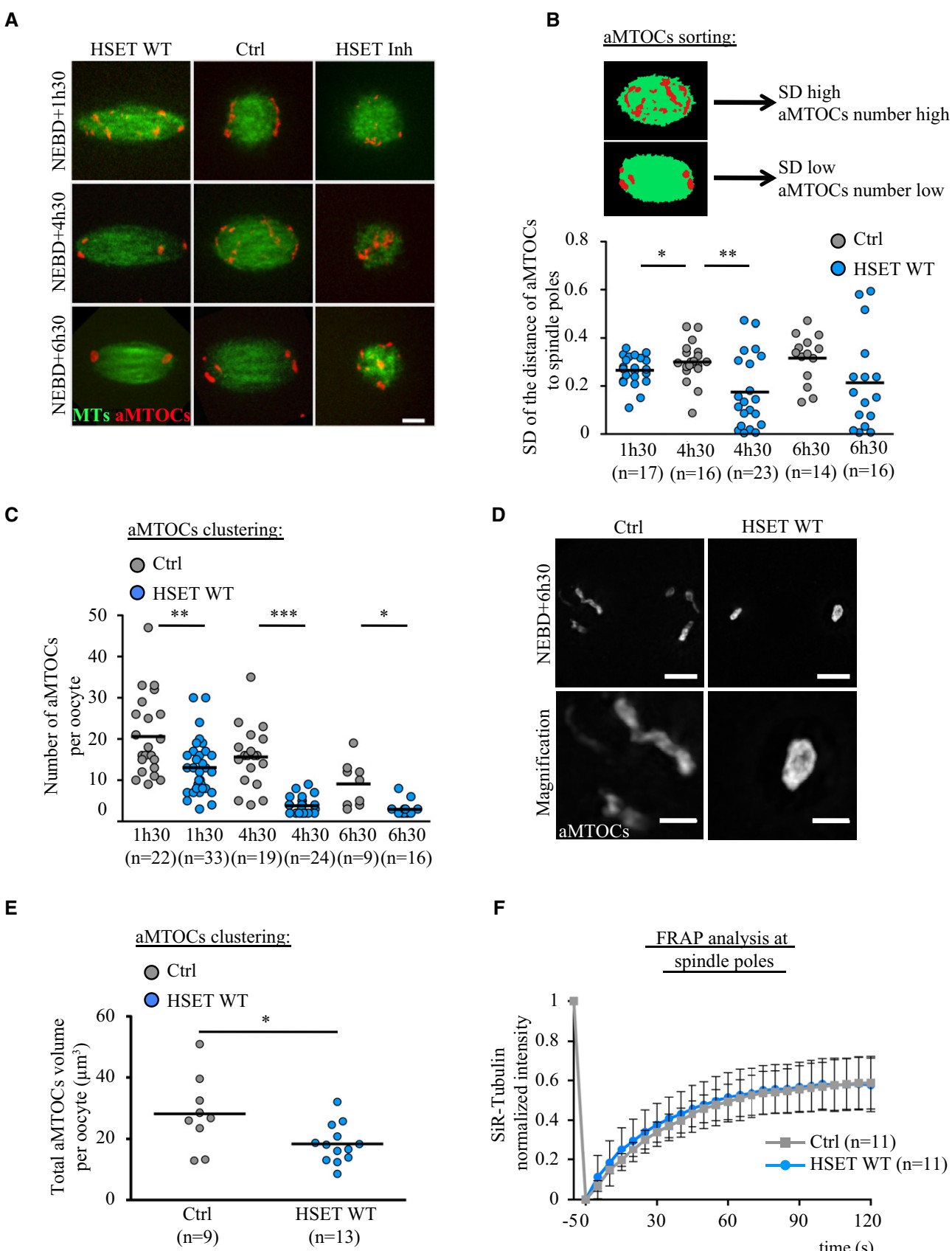

Figure 2.

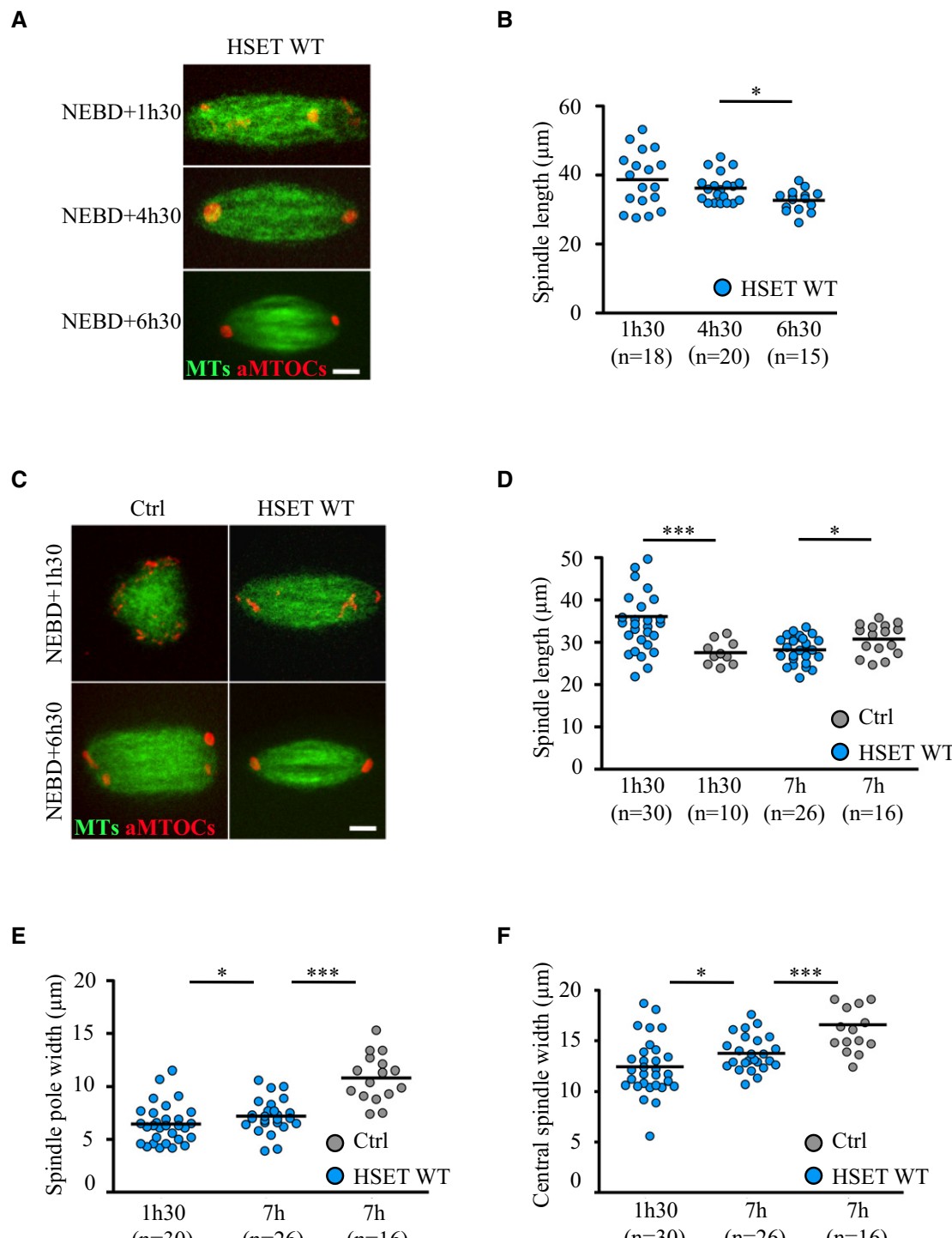

**Figure 3. Turning meiosis I spindles into more mitotic ones.**

A    Spinning disk confocal microscopy images showing spindle region magnifications of oocytes expressing GFP-EB3 (green), mCherry-Plk4 (red), and HSET WT at NEBD+1h30, +4h30, and +6h30. Scale bar 10 μm.

B    Quantification in 3D of spindle length in oocytes expressing HSET WT (blue dots) at NEBD+1h30, +4h30, and +6h30. Each dot represents an oocyte; the number of oocytes analyzed is written in parentheses. Statistical significance of differences is assessed with a Mann–Whitney test: *P-value = 0.011.

C    Spinning disk confocal microscopy images showing spindle region magnifications of oocytes expressing GFP-EB3 (green) and mCherry-Plk4 (red) together with HSET WT (HSET WT, right panels) or not (Ctrl, left panels) at NEBD+1h30 and NEBD+6h30. Scale bar 10 μm.

D–F  Quantification in 3D of spindle length (D) *P-value = 0.022, ***P-value < 0.0001; spindle pole width (E) *P-value = 0.046, ***P-value < 0.0001 and central spindle width (F) *P-value = 0.037, ***P-value < 0.0001 in oocytes overexpressing HSET WT (blue dots) and controls (gray dots) at NEBD+1h30 and NEBD+7h. Each dot represents an oocyte; the number of oocytes analyzed is written in parentheses. Statistical significance of differences is assessed with a Mann–Whitney test.

the spindle poles begin to focus following clustering of the aMTOCs. In oocytes inhibited for HSET by treatment with AZ82, the spindle was not yet bipolar at NEBD+6h30 (Figs 2A and EV3C, right panels). Instead, these spindles remained in a ball-shape, as quantified in Fig EV3D. The diameter of the microtubule mass even decreased slightly between the first and last time points in the HSET-inhibited oocytes (Fig EV3D, purple dots, 25 μm at NEBD+1h30 vs. 22 μm at NEBD+6h30), whereas in control oocytes, the spindle elongated (Fig EV3D, gray dots, 26 μm at NEBD+1h30 vs. 33 μm at NEBD+6h30). Therefore, measurements of aMTOCs sorting and clustering were not relevant in oocytes inhibited for HSET and we focused our analysis on oocytes overexpressing HSET WT where spindle bipolarization is advanced.

We first analyzed aMTOC sorting in controls and HSET WT oocytes (Figs 2A and EV3C, middle and left panels). To do so, the distribution of the aMTOCs was measured in 3D along the long axis of the spindle at the time points where the spindle is bipolar (Fig EV3B, each dot corresponds to one aMTOC, the horizontal axis represents an hemi-spindle from the central spindle to the pole, the distance of aMTOCs to the nearest spindle pole is normalized by the spindle length, and no measurements were conducted at NEBD+1h30 in controls since at that stage spindles are not yet bipolar). In controls, the spindle was bipolar at NEBD+4h30 and the aMTOCs were scattered along the spindle's long axis (Fig EV3B, upper panel, all the gray dots are homogeneously distributed along the hemi-spindle). At NEBD+6h30, the aMTOCs were partially sorted and began to accumulate at spindle poles (Fig EV3B, upper panel, gray dots). We also plotted the standard deviation of the distribution of aMTOCs along the axis of the spindle for each oocyte analyzed (Fig 2B, each dot represents one oocyte). Before aMTOCs are sorted, the standard deviation is high; in contrast, once they are sent to the poles, the standard deviation is low (Fig 2B, scheme). In controls, the difference between NEBD+4h30 and +6h30 was small, highlighting the fact that aMTOC sorting is a long and progressive process (Fig 2B, gray dots). In oocytes overexpressing HSET, the spindle was already bipolar at NEBD+1h30 and aMTOCs were scattered along its long axis (Fig EV3B, lower panel, blue dots are homogeneously distributed along the hemi-spindle), resembling the NEBD+4h30 time point in controls. At NEBD+4h30, the aMTOCs

were partially sorted as indicated by their substantial accumulation at spindle poles (Fig EV3B, lower panel, blue dots), resembling the NEBD+6h30 time point in controls. By NEBD+6h30, aMTOCs were further sorted (Fig EV3B, lower panel, blue dots). The standard deviation of the distribution of aMTOCs along the axis of the spindle for each oocyte showed the same behavior (Fig 2B, blue dots): The standard deviation at NEBD+1h30 in oocytes overexpressing HSET WT was comparable to the standard deviation at NEBD+4h30 in the controls, and at NEBD+4h30 and 6h30, it was smaller than in the controls. Altogether, these results show that aMTOC sorting takes place precociously in oocytes overexpressing HSET WT.

We then analyzed aMTOC clustering in controls and oocytes overexpressing HSET WT (Figs 2A and EV3C, middle and left panels). To do so, the number of aMTOCs per oocyte was counted in 3D (Fig 2C, each dot represents one oocyte). In controls, the number of aMTOCs diminished in parallel with meiosis I progression (Fig 2C, gray dots). This shows that aMTOCs tend to fuse and cluster during meiosis I. In oocytes overexpressing HSET WT, this process started earlier when the spindle bipolarized around NEBD+1h30, as evidenced by a reduced number of aMTOCs (Fig 2C, compare blue and gray dots). Later during meiosis I, the clustering of aMTOCs continued to be enhanced compared to controls (Fig 2C, compare blue and gray dots). Interestingly, aMTOCs were also more compact in oocytes overexpressing HSET WT compared to controls (Fig 2A and D). First, their organization was different: in controls, aMTOCs formed a typical O-shaped structure circumscribing the poles [9], whereas in oocytes overexpressing HSET WT, they formed a single round entity (Fig 2A and D). Second, they occupied a smaller volume as quantified from the N-SIM super-resolution images (Fig 2E). This suggests that HSET may play a role in the spacing of aMTOCs at spindle poles.

We next assessed whether microtubule dynamics was altered in the hyper-clustered spindle poles of oocytes overexpressing HSET WT. To compare microtubule dynamics, we performed FRAP of SiR-Tubulin at spindle poles at NEBD+6h30 (Fig 2F). Essentially identical recovery curves were observed in oocytes overexpressing HSET WT and controls, indicating that microtubule dynamics at spindle poles was similar in the two groups. This strongly suggests that changes in microtubule nucleation or stability are not the root of the difference in spindle pole focusing.

**Figure 4. HSET levels must be tightly gated during early stages of spindle morphogenesis.**

A   Immunofluorescence on fixed control oocytes showing that endogenous HSET (HSET antibody: gray) is present in mouse oocytes from Prophase I and localized on the spindle at NEBD+1h, NEBD+5h, NEBD+8h, metaphase of meiosis II (MII), and in the nucleus 6 h after parthenogenetic activation. Scale bar 10 μm.

B   Endogenous HSET levels progressively increase throughout meiosis I. Endogenous HSET intensity measured for fixed control oocytes in Prophase I, at NEBD+1h, NEBD+5h, NEBD+8h, metaphase of meiosis II (MII), and after activation. Data are represented as mean ± SD. Statistical significance of differences is assessed with a Mann–Whitney test: *$P$-value = 0.017, **$P$-value = 0.0017, ***$P$-value < 0.0001. The ratio of HSET expression between NEBD+8h and NEBD+1h is 1.7, the ratio of HSET expression between NEBD+1h and Prophase I is 1.6, and the ratio of HSET expression between activated and Prophase I oocytes is 1.28.

C   Scheme of the experimental setup for early and late HSET perturbations. DNA is in blue, microtubules in green, aMTOCs in red, NEBD stands for nuclear envelope breakdown.

D   Spinning disk confocal microscopy images showing spindle region magnifications of oocytes expressing HSET WT (late HSET OE, cRNA injected at NEBD+4h, left panel), controls (middle panel) and oocytes inhibited for HSET at NEBD+4h (late HSET Inh, right panel), all imaged at NEBD+6h30. All oocytes express mCherry-Plk4 (red), Ctrl and late HSET Inh oocytes express GFP-EB3 (green), and late HSET OE oocytes express GFP-HSET WT (green). Scale bar 10 μm.

E   GFP-HSET WT total fluorescence intensity measured in the whole cell was assessed after cRNA injection in early overexpression oocytes (cRNA injected in Prophase I, gray bars) and late overexpression oocytes (cRNA injected at NEBD+4h, blue bar). The number of oocytes analyzed is written in parentheses. The total GFP-HSET WT fluorescence intensity for early HSET overexpression oocytes at NEBD+1h is 2.77 ± 1.47 arbitrary units (a.u.) and 2.40 ± 0.97 a.u. at NEBD+7h compared to 3.95 ± 1.70 a.u. for late HSET overexpression oocytes. Standard deviation is plotted on each bar. Statistical significance of differences is assessed with a Mann–Whitney test: not significant (n.s.) $P$-value = 0.408, *$P$-value = 0.048, ***$P$-value < 0.0001.

F, G   Quantification of the spindle length (F) and spindle pole width (G) in late HSET WT overexpression oocytes (cRNA injected at NEBD+4h) and controls at NEBD+6h30. Statistical significance of differences is assessed with a $t$-test: for (F), not significant (n.s.) $P$-value = 0.06; for (G), not significant (n.s.) $P$-value = 0.96. Each dot represents an oocyte, and the number of oocytes analyzed is written in parentheses.

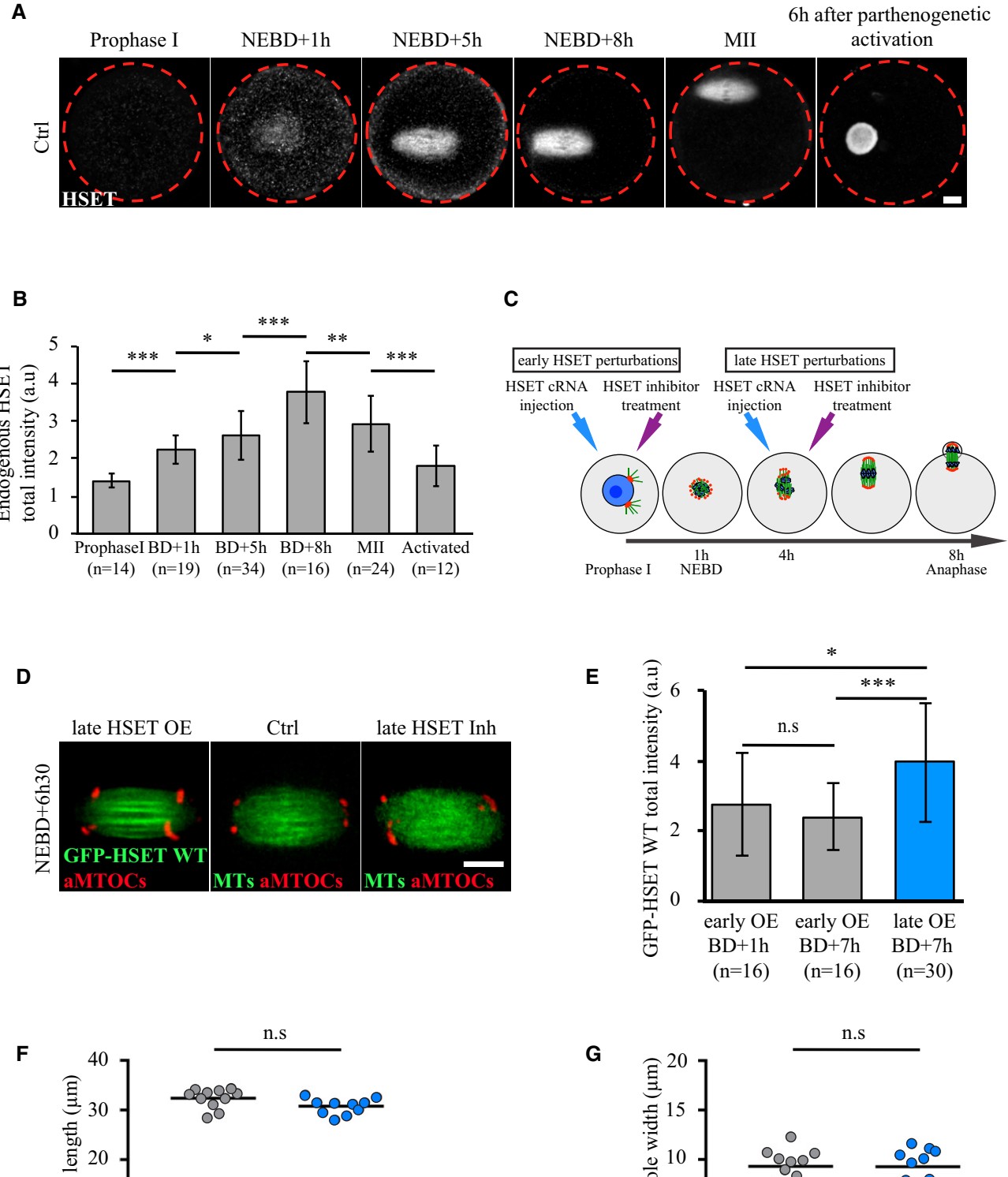

Figure 4.

These results thus show that the timing of spindle morphogenesis is accelerated in oocytes overexpressing HSET WT: Spindle bipolarization is established precociously together with more efficient sorting and clustering of aMTOCs, markers of spindle pole assembly. We then analyzed the impact of accelerated kinetics on spindle shape.

## Shifting meiotic spindle morphology toward mitotic-like morphology

To determine whether accelerating bipolarization and spindle pole formation affected global spindle shape, spindle length, central spindle width, and spindle pole width were measured at the same time points used to analyze aMTOC behavior. In oocytes overexpressing HSET WT, the spindle was already bipolar at NEBD+1h30. Strikingly, the spindle at this stage was extraordinarily long (Fig 3A, live microscopy), with a mean length of 36 μm and reaching a maximum of 54 μm (Fig 3B). As previously shown, this effect of HSET overexpression on spindle length required microtubule sliding [23], as HSET N593K expressing oocytes displayed spindle lengths similar to controls (Fig EV4A). However, in oocytes overexpressing HSET WT, the spindles progressively shortened (Fig 3B) to reach a size comparable to controls by NEBD+7h (Fig 3D).

Even though spindles in oocytes overexpressing HSET WT did recover in length (Fig 3D), we observed significant alterations in their shapes—both the poles and the central region were thinner, and the poles were pointed in contrast to the typical barrel-shape of meiotic spindle poles (Fig 3A and C). To confirm these observations, we measured spindle pole and central spindle widths at NEBD+1h30 and +7h. Oocytes overexpressing HSET WT displayed significantly reduced pole (Fig 3E) and central spindle widths (Fig 3F) compared to controls. In contrast, the spindle pole width of oocytes expressing HSET N593K was the same as controls (Fig EV4B). Thus, HSET levels, likely due to its microtubule-sliding activity, must be tightly regulated for proper spindle architecture in meiosis I.

## HSET level must be tightly gated during early spindle morphogenesis

In an effort to understand how minor deregulation of HSET level impacts the entire process of spindle morphogenesis, endogenous HSET expression levels were measured before, throughout and after meiosis. Levels of HSET were low in Prophase I (1.6 times less than at NEBD+1h, Fig 4A and B). Endogenous HSET levels increased 1.7-fold over the course of 7 h during meiosis I (Fig 4A and B). Interestingly, HSET levels were 1.28 times higher after meiosis (in activated oocytes, mimicking fertilization) compared to Prophase I-arrested oocytes (Fig 4A and B). In addition, HSET was strongly enriched in the female pronucleus after parthenogenetic activation (Fig 4A). Thus, after meiosis, the zygote, comparable in size to the oocyte and similarly devoid of centrioles in rodents, enters the first mitotic division with more HSET than Prophase I-arrested oocytes. Consistently, spindle shape in the zygote is more mitotic-like: elongated, with focused poles [33,34].

Next, we tested whether the alteration of spindle shape observed after HSET WT overexpression was a direct consequence of impairing early spindle morphogenesis via modifying HSET levels prior spindle morphogenesis or was due to the continuous perturbation of HSET levels throughout the first meiotic division. To discriminate between these two hypotheses, HSET levels were modified after early stages of spindle morphogenesis had occurred, namely at NEBD+4h once the spindle was already bipolar (Fig 4C, late HSET perturbations). Spindles were further observed at NEBD+6h30. Spindle shape was comparable to controls—both displaying a typical barrel-shape (Fig 4D, F and G)—following this late HSET increase. This was not due to lower expression levels reached in late versus early perturbations of HSET. Indeed, late injection of GFP-HSET WT allowed the production of levels of HSET that were 1.4 times higher than early injections (Fig 4E, compare gray and blue bars). Thus, the alteration of spindle shape observed after HSET WT overexpression is a direct consequence of impairing early stages of spindle morphogenesis, rather than due to continuous perturbation of HSET levels later during the first meiotic division.

## Mitotic-like spindles display defects in chromosome alignment and segregation

We next asked what were the consequences of forcing a mitotic-like mode of spindle morphogenesis on chromosome alignment and segregation. To answer this question, chromosome behavior was followed in living oocytes. Whereas at NEBD+1h30 in controls, the chromosomes were inside the microtubule ball (Fig 5A, upper left panel), in most oocytes overexpressing HSET WT, the chromosomes were scattered from pole to pole along the extended spindles (Fig 5A, upper right panel). This is in striking contrast to the situation in controls where the microtubule ball elongates

---

**Figure 5.  Early spindle morphogenesis defects induce errors in chromosome alignment and segregation.**

A  Spinning disk confocal microscopy images showing spindle region magnifications of controls (left panel) and oocytes overexpressing HSET WT (right panel) at NEBD+1h30 and +6h30. All oocytes express GFP-EB3 (green) and Histone-RFP (blue). The white asterisk marks a chromosome outside of the metaphase plate. Scale bar 10 μm.

B  Spinning disk confocal microscopy images showing spindle region magnifications of controls (left panel) and oocytes overexpressing HSET WT (right panel) before and after anaphase. Oocytes express GFP-EB3 (green, left panel) or GFP-HSET WT (green, right panel) and Histone-RFP (blue). The white asterisk marks a lagging chromosome in the oocyte before and after anaphase. Scale bar 10 μm.

C  Graph representing the percentage of oocytes with aligned (gray) and not aligned (black) chromosomes before anaphase, quantified for controls (left bar), and oocytes expressing HSET WT (right bar). Statistical significance of differences is assessed with a Fisher test: **P-value = 0.006.

D  Spinning disk confocal microscopy images showing spindle region magnifications of controls (left panel) and oocytes overexpressing HSET N593K (right panel) before and after anaphase. Oocytes express GFP-EB3 (green, left panel) or GFP-HSET N593K (green, right panel) and Histone-RFP (blue). Scale bar 10 μm.

E  Graph representing the percentage of oocytes with aligned (gray) and not aligned (black) chromosomes before anaphase, quantified for controls (left bar) and HSET N593K oocytes (right bar). Statistical significance of differences is assessed with a Fisher test: not significant (n.s.) P-value > 0.99.

**A**

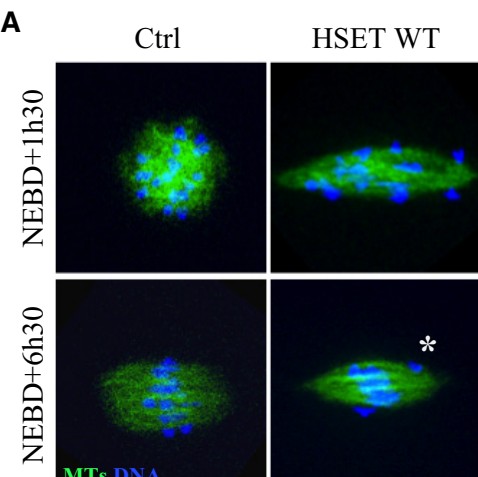

**B**

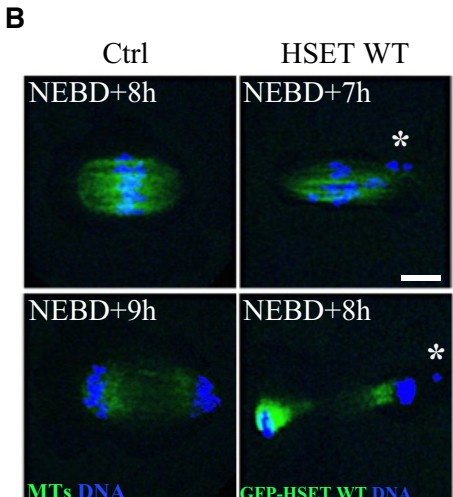

**C**

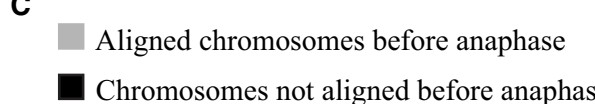

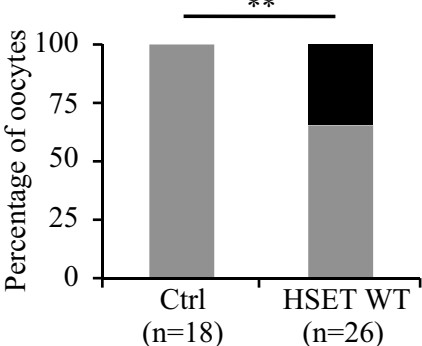

**D**

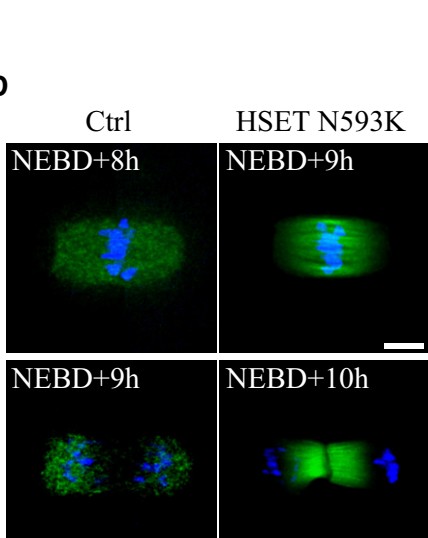

**E**

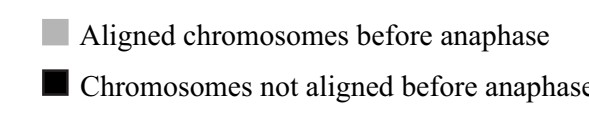

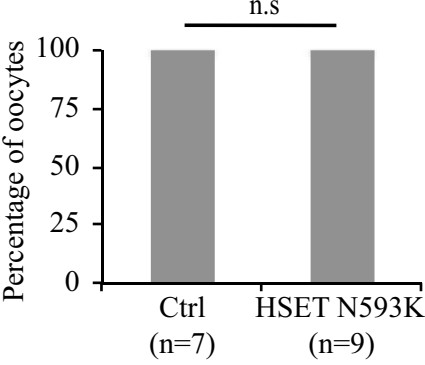

**Figure 5.**

perpendicular to the prometaphase belt, transforming into a barrel-shaped bipolar spindle with the chromosomes gathered in the central region [13,35].

In control oocytes, once the chromosomes have become aligned on the central spindle region to form a metaphase plate, anaphase can occur (Fig 5B, left lower panel). In oocytes overexpressing HSET WT, even when the spindle reached a normal length (Fig 3C and D), the chromosomes remained incompletely aligned on the metaphase plate before anaphase and often presented more than one lagging chromosome outside the metaphase plate (Fig 5A and B, right panels; white asterisks). Indeed, 40% of HSET WT overexpressing oocytes harbored chromosomes that were not aligned before anaphase (Fig 5C, black bar), whereas misalignment was never observed in controls (Fig 5C, gray bar). Despite the presence of misaligned chromosomes, oocytes overexpressing HSET WT nonetheless underwent anaphase, leading to aberrant chromosome segregation (Fig 5B, right lower panel; the white asterisk indicates a chromosome separated from the rest of the main chromosome mass retained in the oocyte after anaphase). Anaphase occurred only with a modest delay (Fig EV5), probably due to the SAC hyposensitivity in meiosis I [36,37].

We then analyzed chromosome alignment in oocytes overexpressing HSET N593K, which in contrast to the HSET WT overexpressing spindles did not bipolarize quickly. In HSET N593K overexpressing oocytes, spindles did not skip the microtubule ball stage (Fig 1E, lower panel) and did not display the mitotic-like spindle shape of oocytes overexpressing HSET WT (Figs 1E and 5D). In addition, the chromosomes were properly aligned on the metaphase plate before anaphase occurred (Fig 5D, right panel and E). These observations further re-enforce the view that chromosome alignment/segregation defects displayed by oocytes overexpressing HSET WT are not a consequence of HSET WT overexpression *per se* but rather a consequence of a shift toward a mitotic mode of spindle morphogenesis.

## Discussion

We have shown that HSET levels must be tightly gated during meiosis I and that deregulation of HSET amount can be used as a tool to force spindle morphogenesis to be more mitotic-like in several aspects: accelerated kinetics of spindle bipolarization and spindle pole assembly coupled with focused poles. Interestingly, this mild HSET overexpression may be comparable to the physiological transition from meiosis I to meiosis II and beyond, that is, first zygotic mitosis. This shift toward mitotic-like spindle morphogenesis is sufficient to severely impair chromosome alignment.

Importantly, late HSET perturbations (overexpression or inhibition) have no effect on spindle shape. This strongly argues that the mitotic-like spindle shape observed after perturbing HSET levels early on are not due to a late and cumulative effect of HSET overexpression throughout meiosis I, but are strictly attributable to increasing HSET levels during early stages of meiosis I. In addition, we did not observe similar defects in spindle bipolarization, pole assembly or chromosome alignment in oocytes overexpressing HSET N593K that can cross-link but not slide microtubules [23]. Interestingly, mouse zygotes also lack centrioles, yet assemble spindles with mitotic characteristics: rapid bipolarization and focused pole

formation [33,34]. Accordingly, we show that zygotes enter mitosis with more HSET than oocytes when they resume meiosis. This mitotic-like mode of spindle morphogenesis in the absence of centrioles does not create chromosome alignment abnormalities in zygotes, as it does in oocytes. However, chromosome properties are different between meiosis and mitosis. Meiosis I is peculiar since homologous chromosomes linked by chiasmata progressively align and biorient on the metaphase plate, instead of single chromosomes as in mitosis. Thus, the volume, shape, and occupancy of the objects (the chromosomes) moving toward the metaphase plate are strikingly different in meiosis, and chromosomes are active participants in meiotic spindle assembly [38]. We propose that early overexpression of HSET WT accelerates spindle bipolarization through increased microtubule sliding, skipping the microtubule ball stage, and thus scattering the chromosomes along the spindle axis. Although the spindle recovers in length as meiosis I progresses, its shape remains distorted, harboring mitotic-like pointed poles instead of the classical barrel-shaped meiotic spindle. The microtubule ball stage could serve as a chromosome shepherd to avoid precocious chromosome scattering. This strategy is used by starfish oocytes where an actin fishnet that forms at meiosis resumption gathers the chromosomes, which dispersed throughout the volume of the large nucleus [39]. It is likely that for meiotic spindles assembled "inside-out", the initial steps have to be precisely controlled in order to prevent chromosome defects that could persist throughout meiosis I.

We were surprised that these early spindle defects were not fully rescued over the extremely long duration of meiosis I. In particular, more highly focused poles were shown to be associated with fewer chromosome alignment defects in meiosis when the microtubule ball stage was not skipped [40]. One possibility is that, when bypassing the microtubule ball stage, chromosomes are quickly scattered over a long distance and those located near the poles never become aligned on the metaphase plate. Indeed, polar chromosomes can be found occasionally in unmanipulated oocytes and usually are not able to gather on the metaphase plate before anaphase [35,41]. Microtubule dynamics is the same at hyperfocused (HSET WT OE) and normal barrel-shaped spindle poles, and the total amount or the density of microtubules might be reduced at poles (spindle pole width is significantly reduced in HSET WT overexpressing oocytes compared to controls). Taken together, this might impair the efficiency of capturing and aligning chromosomes that have been lost early on at the poles, a phenomenon that occurs more often in HSET WT overexpressing oocytes because of the early chromosome scattering.

In conclusion, forcing meiosis I spindle morphogenesis to be more mitotic-like leads to chromosome alignment abnormalities that cannot be fully reversed. In an unexpected manner, the unusual length of meiosis I (8 h) is not sufficient to correct early spindle morphogenesis defects, contributing to chromosome misalignment and segregation. This could be relevant to other systems as well, spindle formation being even slower in human oocytes, taking ~15 h [18]. Avoiding a mitotic-like mode of spindle morphogenesis could be one reason why most oocytes lose canonical centrosomes. It is thus possible that mouse oocytes, and maybe also human oocytes, eliminated canonical centrosomes to prevent a mitotic-like mode of spindle assembly during meiosis I, thereby to safeguarding against further increases in aneuploidy levels, already high during this specific division in these species [42].

# Materials and Methods

### Oocyte collection and culture

Ovaries were collected from 11-week-old OF1 (wt) female mice. Fully-grown oocytes were extracted by shredding the ovaries [43] and then releasing the germ cells in M2 + BSA medium supplemented with 1 µM milrinone to block and synchronize them in Prophase I of meiosis [44]. Meiosis resumption was triggered by transferring oocytes into milrinone-free M2 + BSA medium. All live-culture and imaging were carried out under oil at 37°C.

### Oocyte activation

Oocytes in metaphase of meiosis II were incubated 2 h in M2 + BSA medium lacking $CaCl_2$ and supplemented with 10 mM $SrCl_2$. Activated oocytes were then cultured in M2 + BSA medium for 6 h until pronuclear formation.

### Constructs

hHSET WT and hHSET N593K were subcloned from plasmids provided by Claire E. Walczak (Indiana University, USA) into a pRN3 plasmid suitable for *in vitro* cRNA transcription. The hHSET WT and N593K expressing plasmids were amplified using One shot Top 10 competent bacteria (Invitrogen), subsequently extracted and purified using mini and midi prep kits (Qiagen).

We used the following constructs: pRN3-GFP-hHSET, pRN3-hHSET, pSpe3-GFP-hHSET-N593K, pRN3-GFP-EB3 [16,17], pRN3-Histone(H2B)-RFP [16,17], pCS2-mCherry-Plk4 [45].

### In vitro transcription of cRNAs and microinjection

Plasmids were linearized using appropriate restriction enzymes. cRNAs were synthesized with the mMessage mMachine kit (Ambion) and subsequently purified using the RNAeasy kit (Qiagen). Their concentration was measured using NanoDrop 2000 from ThermoScientific. cRNAs were centrifuged at 4°C during 45 min prior to microinjection into the cytoplasm of oocytes blocked in Prophase I in M2 + BSA medium supplemented with 1 µM milrinone at 37°C. cRNAs were microinjected using an Eppendorf Femtojet microinjector [46]. After microinjection, cRNA translation was allowed for 1 h, and oocytes were then transferred into milrinone-free M2 + BSA medium to allow meiosis resumption and meiotic divisions. For Fig 4C–G, late HSET OE oocytes were microinjected at NEBD+4h.

### HSET WT and HSET N593K overexpression experiments

Oocytes were microinjected with 150 ng/µl of hHSET WT cRNAs or 250 ng/µl of GFP-hHSET WT cRNAs. We have observed that this is the optimal concentration to detect interpretable phenotypes whereas lower or higher concentrations gave, respectively, no phenotypes or induced spindle collapse and mono-aster formation. Oocytes were microinjected with 250 ng/µl of GFP-hHSET N593K. After microinjection, cRNA translation was allowed for 1 h and oocytes were transferred into milrinone-free M2 + BSA medium to allow meiosis resumption.

### Drug treatment

The AZ82 inhibitor of HSET was a gift from AstraZeneca (USA) [29–30]. AZ82 was stored diluted in DMSO at 100 µM and further diluted in M2 medium to a final concentration of 10 µM. Control experiments were done in M2 + BSA medium supplemented with equivalent concentrations of DMSO.

The CW069 inhibitor of HSET was a gift from Fanni Gergely (Cancer Research UK Cambridge Institute, UK) [31]. CW069 was stored diluted in DMSO at 100 µM and then diluted in M2 medium at a final concentration of 25 µM. Control experiments were done in M2 + BSA medium with equivalent concentrations of DMSO. It had been previously shown that a concentration of 1/100 of DMSO in M2 medium does not perturb oocyte maturation [47].

### Live imaging and SIM super-resolution microscopy

Spinning disk movies were acquired using a Plan-APO 40×/1.25NA objective on a Leica DMI6000B microscope enclosed in a thermostatic chamber (Life Imaging Service) equipped with a CoolSnap HQ2/CCD camera coupled to a Sutter filter wheel (Roper Scientific) and a Yokogawa CSU-X1-M1 spinning disk. Metamorph Software (Universal Imaging) was used to collect data.

For SIM super-resolution microscopy of aMTOCs, image acquisition was performed in 3D SIM mode, with a N-SIM Nikon microscope (Nikon Imaging Centre @ Institut Curie-CNRS) before image reconstruction using the NISElements software [48]. The system is equipped with an APO TIRF 100× 1.49NA Oil Immersion, a laser illumination (488 nm at 200 mW and 561 nm at 100 mW), and an EMCCD DU-897 Andor camera.

### Immunofluorescence

After *in vitro* culture of oocytes, their zona pellucida was removed by incubation in acid Tyrode's medium (pH = 2.3). Prophase I-arrested oocytes were incubated in M2 + BSA medium supplemented with 0.4% pronase to remove the zona pellucida.

To visualize aMTOCs by SIM, oocytes were fixed 30 min at 30°C in 4% formaldehyde at NEBD+6h30 on coverslips treated with gelatin and polylysine. Permeabilization was achieved by incubating oocytes in 0.5% Triton X-100 in PBS for 10 min at room temperature. Mouse anti-pericentrin antibody (BD Transduction Laboratories) was used at 1:2,000. As secondary antibody, anti-mouse Cy3 (Molecular Probes) was used at 1:600. Slides were mounted in ProLong Gold.

To visualize endogenous HSET and exogenous microinjected hHSET, oocytes were fixed 30 min at 30°C in 3.7% formaldehyde and permeabilized 10 min at room temperature in 0.25% Tween-20–PBS. The HSET antibody was a gift from Renata Basto (Curie Institute, Paris, France). As secondary antibody, anti-rabbit Cy2 (Molecular Probes) was used at 1:200. Chromosomes were stained with Prolong-DAPI (10 µg/ml final DAPI).

### FRAP analysis

Images were acquired using a Plan-APO 60×/1.4NA objective on a Ti Nikon microscope enclosed in a thermostatic chamber (Life Imaging Service) equipped with a Flash4.0 V2 CMOS camera

(Hamamatsu) coupled to a Yokogawa CSU-X1 spinning disk. Metamorph Software (Universal Imaging) was used to collect data. All oocytes expressed SiR-Tubulin (from Spirochrome reference SC002, used at 0.1 μM). For all oocytes, an identical region of interest (diameter of 5 μm) was bleached at spindle poles. Images were acquired every 5 s for 125 s. One image was taken before bleaching. The SiR-Tubulin fluorescence intensity quantification was performed using the Metamorph software (Universal Imaging). Normalization of the measured fluorescence intensities was performed using the Microsoft Excel software. As the expression levels of SiR-Tubulin vary from one experiment to another, the signal intensity was normalized so that the prebleached value was 1 and the value at the first time point after bleaching was 0.

## Quantifications

Metamorph (Universal Imaging), Imaris (Oxford Instruments), and Fiji (NIH) software were used to analyze and process data.

(i)    The timing of spindle bipolarization was measured on oocytes expressing GFP-EB3 or SiR-Tubulin (from Spirochrome reference SC002, used at 0.1 μM) using Metamorph software; bipolarity was scored when two poles were distinguishable (see Fig 1A–F).

(ii)    For endogenous and exogenous HSET intensity measurements on fixed samples, HSET intensity was measured inside a circle of a fixed size having the mean diameter of all oocytes (see Fig EV1C and D).

(iii)    The GFP-HSET WT intensity measurements 3 h after cRNA injection (see Fig 4E) were performed using Metamorph software. After background subtraction, the total fluorescence intensity was measured inside a circle of a fixed size having the mean diameter of all oocytes.

(iv)    Chromosome alignment before anaphase was measured on oocytes expressing Histone-RFP using Metamorph software (see Fig 5).

(v)    For the aMTOCs 3D analysis (see Fig 2A–C), the input data consist of multichannel *Z*-stack images from spinning disk microscopy, containing bright-field, GFP and RFP channels. A homemade plug-in was developed for ImageJ/Fiji software to analyze aMTOCs position within the spindle. This 3D_Spindle_Analysis plug-in is available at https://github.com/pmailly/3D_Spindle_Analysis. Oocyte boundaries were first extracted with variance filter and triangle method for thresholding from the bright-field channel and used to crop the image in the other two channels. Spindle (GFP channel) was first filtered using 3D Gaussian filter (radius = 2) to reduce noise and then thresholded using MaxEntropy method. aMTOCs (RFP channel) were first filtered using difference of Gaussians (GDSC libraries from Alex Herbert, University of Sussex) to increase spot-like signals, then thresholded using MaxEntropy method. For each channel, 3D objects were segmented using the 3D ImageJ suite [49]. The spindle poles positions were computed as the extremities of the larger diameter of the object (Feret diameter). For each aMTOCs, minimum distances to poles, and border distances to the spindle were computed.

(vi)    The spindle length, central spindle width, and spindle pole measurements were performed in 3D using Imaris software (see Figs 3 and EV3D). The spindle poles positions were considered as being the extremities of the larger diameter of the spindle, and spindle length was measured as the distance between poles.

(vii)    The aMTOCs volume measurements were performed in 3D using Imaris software (see Fig 2D and E). The input data consist of SIM super-resolution microscopy acquisitions performed in 3D SIM mode, and the total volume of aMTOCs per oocyte was measured.

## Statistical analysis

Experiments were repeated at least three times, and a sample of sufficient size was used. The statistical analysis was performed using GraphPad Prism version 7.00 for MacOS, GraphPad Software, La Jolla, California, USA, www.graphpad.com. For comparisons between two groups, the normality of the variables was checked (D'agostino-Pearson normality test) and parametric Student's *t*-tests (with Welch correction when indicated) or non-parametric comparison tests were performed with a confidence interval of 95%. For chromosome alignment experiments, repartitions were analyzed for statistical significance using Fisher's test used with a confidence interval of 95%. All error bars are expressed as standard deviation (SD). Values of $P < 0.05$ were considered significant. In all figures, * corresponds to a *P*-value $< 0.05$, ** to a *P*-value $< 0.005$, *** to a *P*-value $< 0.0001$. n.s.: not statistically significant.

**Expanded View** for this article is available online.

## Acknowledgements
We thank AstraZeneca (USA) for the gift of AZ82, Fanni Gergely (Cancer Research UK Cambridge Institute, UK) for the gift of CW069, and Claire E. Walczak (Indiana University, USA) for the gift of plasmids. We thank Renata Basto (Curie Institute, Paris, France) for sharing the unpublished HSET antibody. We thank Lucie Sengmanivong for her precious help using the N-SIM Nikon microscope from the Nikon Imaging Centre @ Institut Curie-CNRS, member of the France-BioImaging national research infrastructure for confocal microscopy. We thank Christophe Klein (Centre de Recherche des Cordeliers-INSERM, Paris, France) for his help with the FRAP analysis. We also thank Hugh J. Clarke (McGill University, Montreal, Canada) for critical reading of the manuscript, the members of the Verlhac/Terret team for helpful discussions, and Anne Dumas (Collège de France, Paris, France) for her help with MTA writing. We thank the "Fondation Bettencourt Schueller". This work was supported by grants from the Fondation pour la Recherche Médicale (FRM label DEQ20150331758 to MHV), from the Fondation ARC (PJA20131200412 to MET), and from Inca (PLBIO 2016-270-TRAN). This work has received support under the program "Investissements d'Avenir" launched by the French Government and implemented by the ANR, with the references: ANR-14-CE11/DIVACEN/MHV, ANR-10-LABX-54 MEMO LIFE, ANR-11-IDEX-0001-02 PSL* Research University.

## Author contributions
IB, AK, M-HV, and M-ET designed the experiments, interpreted the results, and wrote the manuscript. IB, IQ, AK, and M-ET carried out the experiments. TB and PM designed the Fiji plug-in. M-ET and M-HV supervised the project.

## Conflict of interest
The authors declare that they have no conflict of interest.

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
