## [Review Process File · EMBO Reports]

Shifting meiotic to mitotic spindle assembly in oocytes disrupts chromosome alignment

Isma Bennabi, Isabelle Quéguiner, Agnieszka Kolano, Thomas Boudier, Philippe Mailly, Marie-Hélène Verlhac and Marie-Emilie Terret

Review timeline:	Submission date:	23 September 2017
	Editorial Decision:	26 September 2017
	Revision received:	21 November 2017
	Editorial Decision:	11 December 2017
	Revision received:	11 December 2017
	Accepted:	13 December 2017

Editor: Achim Breiling

Transaction Report: This manuscript was transferred to *EMBO reports* following review at *The EMBO Journal*

1st Editorial Decision

26 September 2017

Thank you for the transfer of your research manuscript to EMBO reports. I now went through the referee reports from The EMBO Journal.

All referees acknowledge the potential interest of the findings. Nevertheless, they have raised a number of concerns and suggestions to improve the manuscript, or to strengthen the data and the conclusions drawn. As the reports are below, I will not detail them here. As EMBO reports emphasizes novel functional over detailed mechanistic findings, we will not require further mechanistic insight, or modeling studies (referee #3, point 2).

However, we ask you to address all the concerns of ref. #2 and point 1 of ref #3, and the points listed by referee #3 as "further concerns". I think it will also be useful to have the revised manuscript corrected by a native speaker before re-submission. Given the constructive referee comments, we would like to invite you to revise your manuscript with the understanding that the referee concerns must be fully addressed in the revised manuscript (as detailed above) and in a complete point-by-point response.

Acceptance of your manuscript will depend on a positive outcome of a second round of review, based on the re-evaluation by ref #2 and #3. It is EMBO reports policy to allow a single round of revision only and acceptance or rejection of the manuscript will therefore depend on the completeness of your responses included in the next version of the manuscript.

Revised manuscripts should be submitted within three months of a request for revision; they will otherwise be treated as new submissions. Please contact us if a 3-months time frame is not sufficient for the revisions so that we can discuss the revisions further.

Please refer to our guidelines for preparing your revised manuscript:

<http://embor.embopress.org/authorguide#manuscriptpreparation>

Supplementary/additional data: The Expanded View format, which will be displayed in the main HTML of the paper in a collapsible format, has replaced the Supplementary information. You can submit up to 5 images as Expanded View. Please follow the nomenclature Figure EV1, Figure EV2 etc. The figure legend for these should be included in the main manuscript document file in a section called Expanded View Figure Legends after the main Figure Legends section. Additional Supplementary material should be supplied as a single pdf labeled Appendix. The Appendix includes a table of content on the first page, all figures and their legends. Please follow the nomenclature Appendix Figure Sx throughout the text and also label the figures according to this nomenclature. For more details please refer to our guide to authors.

Important: All materials and methods should be included in the main manuscript file.

Regarding data quantification and statistics, can you please specify the number "n" for how many experiments were performed, the bars and error bars (e.g. SEM, SD) and the test used to calculate p-values in the respective figure legends? This information must be provided in the figure legends. Please provide statistical testing where applicable.

We now strongly encourage the publication of original source data with the aim of making primary data more accessible and transparent to the reader. The source data will be published in a separate source data file online along with the accepted manuscript and will be linked to the relevant figure. If you would like to use this opportunity, please submit the source data (for example scans of entire gels or blots, data points of graphs in an excel sheet, additional images, etc.) of your key experiments together with the revised manuscript. Please include size markers for scans of entire gels, label the scans with figure and panel number, and send one PDF file per figure or per figure panel.

I look forward to seeing a revised version of your manuscript when it is ready. Please let me know if you have questions or comments regarding the revision.

REFEREE REPORTS

Referee #1:

Usually animal oocytes do not present canonical centrosomes and therefore the spindle is formed from multiple acentriolar MTOCs. In this article, the authors investigate the effect of overexpressing a member of kinesin-14 family, HSET, that has microtubule crosslinking and sliding activity. They observe that overexpression leads to an accelerated spindle bipolarization and clustering of MTOCs at more focused spindle poles. In addition the authors observe chromosome misalignment and segregation defects in HSET overexpressing oocytes.

This study demonstrates that HSET overexpression perturbs spindle assembly and function during meiosis I, in a manner expected for a minus-end directed crosslinking motor. Unfortunately the study focuses mostly on a description of this overexpression phenotype, rather than trying to elucidate the physiological role that HSET may play during meiosis and explore the mechanism behind the functional defects. As HSET overexpression has been shown to affect a number of other mitotic regulators in the past, it is quite likely that it will have a number of indirect rather than direct effects and it is unclear if any of the observed effects are physiologically relevant for meiosis.

Rather than doing this mechanistic work, the authors claim that their overexpression constitutes a "switch to a more mitotic-like spindle" and that they can therefore draw general conclusions about why oocytes normally have less focused and centriole free spindles. This part of the study is

quite weak and purely speculative. Many years of research have already demonstrated that the mechanisms of mitotic and meiotic spindle assembly differ considerably in several molecular aspects: presence or absence of centrioles, the contribution of chromatin-mediated microtubule nucleation (Ran GTP and CPC pathways), mechanisms of chromosome congression and alignment, structure and nature of kinetochore, composition and functionality of M-phase checkpoints, mechanism of physical separation of chromosomes in anaphase, etc.. Assuming that the artificial overexpression of HSET alone can transform the meiotic to mitotic-like spindle apparatus and function is very speculative and the general conclusions the authors attempt to draw here are not substantiated by the data.

In addition, there are a number of technical concerns, regarding the control and measurements of expression levels of HSET, the validity of the image analysis, excluding perturbing effects of the live cell fluorescent marker that are used and a lack of clear figure labeling, referencing and technical description that are much below the level of what I would expect to see in EMBO Journal. Finally, the manuscript is overall poorly written, has several occurrences of unfinished or copied sentences, incorrect use of English language and is not well-organized. After carefully addressing these more technical and organizational issues, this could be a useful descriptive study of an overexpression phenotype in a more specialized journal.

Referee #2:

This study examines the effect of kinesin-14 overexpression and inhibition on meiotic spindle assembly and chromosome segregation in mouse oocytes. The main conclusion is that too much kinesin-14 dependent microtubule sliding in the early phase of meiotic spindle assembly causes spindles to bipolarize too quickly which leads to chromosome misalignment and segregation errors during anaphase. The study impresses by high quality experiments and careful and detailed quantitative analysis of the observations. It is a beautiful quantitative cell biology study. The results are novel, very interesting and significant. Overall the study is very well presented. An excellent manuscript.

This reviewer has only two major issues:

(1) Results: The authors perform a control experiment with a kinesin-14 mutant that is expected to crosslink microtubules but to lack the ability to slide microtubules. This is a nice experiment allowing a straight-forward conclusion about the importance of kinesin-14 motility for the observed effects, provided localization to the spindle is similar as for the wildtype. The authors state that this is the case, and cite Figures EV1B and 1E to support their claim. But it seems that Fig. 1E shows labelled microtubule fluorescence in the presence of the mutant (that is apparently not visualized) and Figure EV1B wildtype GFP-HSET fluorescence. A quantification as performed for the wildtype in Figure EV1D seems to be missing for the mutant, apparently leaving the claim unsupported.

(2) Discussion: The most significant part of the Discussion seems to be the proposal that centrosomes were lost in oocytes to avoid fast mitosis-like spindle bipolarisation to allow for enough time for chromosome alignment. In a way that is a re-statement of the major result. What could turn this summary into a discussion would be a proposed explanation why mitosis-style bipolarization works for mitosis, but not for meiosis, i.e. what's the specific difference between meiosis and mitosis - probably related to chromosome properties - that requires such different mechanisms.

Minor issues

- (3) It is not always clear from the main text and the figure legends how experiments were performed, i.e. which molecules were labelled for microscopy, whether live or fixed (immunostained) specimen were observed and which type of microscopy was used (confocal, SIM?). It would be useful to have this information in each figure legend.
- (4) This reviewer found it odd that the data of Figure EV3B were presented in an accompanying figure, although they very directly show main results, whereas the results of the further analysis were presented in the main figure in a somewhat wordy manner. Can probably be streamlined.
- (5) Figures 3B and D seem to show more or less the same information - really necessary to show both?
- (6) Methods: The description of the methods used for quantifications is rather brief. Some more detail or a link to the used scripts would be desirable.
- (7) Language: Abstract: 'extended and reversed engineered process' - unclear term; page 6: 'plasticity' - definition of what this means in this context is missing = unnecessary; some typos (e.g. Methods: 'Suted filter sheel')

Referee #3:

Bennabi et al aim at analyzing further insights on meiotic spindle assembly in female oocytes. They use the established mouse oocyte system, which allows following the process in real time in a mammalian model. Several studies in recent years have highlighted the kinetic, conceptual and some molecular differences of mitotic vs meiotic spindle formation.

While centrosomes in somatic cells ensure rapid microtubule assembly and spindle pole formation, the absence of centrosomes in vertebrate oocytes necessitates a rather elongated "inside-out" spindle formation mechanism. In mouse oocytes, several non-centrosomal MTOCs are organised into a spindle pole by the action of microtubule-associated and motor proteins finally leading to spindle pole formation.

The authors come up with the intriguing idea to switch meiotic spindle formation into a more mitotic-like pathway using overexpression of the minus-end directed kinesin HSET/KifC5b. Indeed, increasing the levels of HSET leads to faster pole formation in mouse oocytes, somewhat reminiscent of rapid pole formation in mitotic cells. At the same time, the authors document that premature bipolarity gained by mild HSET overexpression compromises proper spindle function.

The data presented here are based on very challenging yet technically excellent experiments. I also have no doubt that this represents a potentially very interesting observation on meiotic spindle formation. However, the work does not unequivocally demonstrate the biological significance of the role of HSET in shifting the system from meiotic to mitotic behaviour. Therefore, due to my opinion, the work is not suitable for publication in EMBO J.

My major concern is twofold:

1. The authors interpret accelerated pole formation after HSET overexpression as a shift towards "mitotic" behaviour. Such a shift would certainly be interesting as it could allow drawing conclusions about the driving force for the evolution of meiotic spindle formation. However, this is very difficult to recapitulate from the data presented here. I do concede that the HSET overexpression situation is well characterised but it is unclear to me if this situation really reflects a sort of mitotic pathway. Is there any evidence that levels/activity of HSET (e.g. compared to MT mass) are higher in somatic cells entering mitosis than in oocytes entering meiosis I? How is HSET regulated before meiosis and after fertilization?

2. Apart from the fact that its MT sliding activity is required for premature MTOC clustering, we do not really understand what increased HSET levels do at the molecular level / MT organization level and why the rather mild increase dramatically changes spindle formation. Modeling may be required to understand what is going on. Is it possible that other kinesin-14 family members may act similarly?

Further concerns:

The HSET overexpression ratios are defined to be 1.6 while 4.2 fold more localizes to the spindle. How are the levels regulated and is there an explanation for the overrepresentation at the spindle? Is it just the mass of MT forming, or is there evidence for an additional level of regulation?

Figure/graph EV2C does not display errors bars although statistical data is presented.

Figure 4 shows endogenous HSET behaviour during MI "accumulating" 1.7 fold with time. I guess this refers to the overall HSET levels? It seems that spindle localisation is stronger. Could this be quantified as well? How is the increased binding interpreted?

Given the importance of the results on chromosome segregation, these results should be more clearly quantified.

Fig. 5C: do we see 18 vs. 26 spindles and respective alignment defects (0 in control, 9 or 10 after HSET wt injection?). Wouldn't this result be the quantification matching the defect shown in Fig. 5A?

Is it correct: the quantification means 15/16 of 26 spindles organized their chromosomes well in metaphase; do they still show chromosome segregation errors, like shown in Fig. 5B?

Several spelling and grammar errors need to be corrected.

1st Revision - authors' response

21 November 2017

Referee#1:

We are puzzled by this report, which we view as unnecessarily aggressive and not constructive. Indeed, this review crucially lacks precise examples and recommendations. The referee does not suggest any specific experiments.

Usually animal oocytes do not present canonical centrosomes and therefore the spindle is formed from multiple acentriolar MTOCs. In this article, the authors investigate the effect of overexpressing a member of kinesin-14 family, HSET, that has microtubule crosslinking and sliding activity. They observe that overexpression leads to an accelerated spindle bipolarization and clustering of MTOCs at more focused spindle poles. In addition the authors observe chromosome misalignment and segregation defects in HSET overexpressing oocytes.

This study demonstrates that HSET overexpression perturbs spindle assembly and function during meiosis I, in a manner expected for a minus-end directed crosslinking motor. Unfortunately the study focuses mostly on a description of this overexpression phenotype, rather than trying to elucidate the physiological role that HSET may play during meiosis and explore the mechanism behind the functional defects. As HSET overexpression has been shown to affect a number of other mitotic regulators in the past, it is quite likely that it will have a number of indirect rather than direct effects and it is unclear if any of the observed effects are physiologically relevant for meiosis.

As you understood it clearly, we never intended to study the role of HSET in meiosis. We also think that the mechanistic underpinning of the phenotype (indirect or direct effects) is of secondary importance here. HSET function was addressed in more suitable model systems such as *Xenopus* egg extracts, amenable to biochemistry. Here, HSET deregulation is used only as a tool to shift meiotic spindle morphogenesis towards a more mitotic one. When we say “more mitotic-like spindle”, we do not mean mitotic strictly speaking, since no centrioles were added back into oocytes. But it is mitotic-like in the way the spindle is assembled: the two spindle poles are formed first, they are focused and not barrel-shaped, bipolarization is precocious compared to meiotic spindle assembly, and the spindle skips the microtubule ball stage. We would be happy to use a better formulation in case the referee has one to suggest.

Rather than doing this mechanistic work, the authors claim that their overexpression constitutes a "switch to a more mitotic-like spindle" and that they can therefore draw general conclusions about why oocytes normally have less focused and centriole free spindles. This part of the study is quite weak and purely speculative. Many years of research have already demonstrated that the mechanisms of mitotic and meiotic spindle assembly differ considerably in several molecular aspects: presence or absence of centrioles, the contribution of chromatin-mediated microtubule nucleation (Ran GTP and CPC pathways), mechanisms of chromosome congression and alignment, structure and nature of kinetochore, composition and functionality of M-phase checkpoints, mechanism of physical separation of chromosomes in anaphase, etc.. Assuming that the artificial overexpression of HSET alone can transform the meiotic to mitotic-like spindle apparatus and function is very speculative and the general conclusions the authors attempt to draw here are not substantiated by the data.

The part of the manuscript described by the Referee as speculative is the discussion. In our opinion, it is precisely a section of the manuscript where results should not only be summarized but also discussed, and where hypothesis, even speculative, can be formulated. It is worth noting that Referee 2 views our discussion as being not enough speculative.

In addition, being one of the pioneer lab working on these issues in mouse oocytes (see references of papers from the lab below), we are certainly aware that the mechanisms of mitotic and meiotic spindle assembly differ in several molecular aspects. Additionally, we wrote an invited review for The Journal of Cell biology last year on this topic. Again, we never wrote in the paper that overexpression of HSET alone can transform the meiotic to mitotic-like spindle apparatus and function (see our answer to the previous point).

In addition, there are a number of technical concerns, regarding the control and measurements of expression levels of HSET, the validity of the image analysis, excluding perturbing effects of the live cell fluorescent marker that are used and a lack of clear figure labeling, referencing and technical description that are much below the level of what I would expect to see in EMBO Journal. Finally, the manuscript is overall poorly written, has several occurrences of unfinished or copied sentences, incorrect use of English language and is not well-organized. After carefully addressing these more technical and organizational issues, this could be a useful descriptive study of an overexpression phenotype in a more specialized journal.

Concerning the last paragraph, the referee raises plenty of points:

- “Technical concerns regarding the control and measurements of expression levels of HSET”.
- Could the referee be more specific and indicate what are his technical concerns, for which figure?
- “Validity of the image analysis”. Could the referee be more precise, especially in terms of figure, and voice his concerns clearly?
- “Lack of clear figure labeling”. Could the referee indicate which figure he/she is refers to?
- “Referencing and technical description are much below the level of what I would expect to see in EMBO Journal”. Can the referee develop and give precise examples? This negative opinion is in

clear contradiction with Referee 2, who states that “The study impresses by high quality experiments and careful and detailed quantitative analysis of the observations. It is a beautiful quantitative cell biology study. The results are novel, very interesting and significant. Overall the study is very well presented. An excellent manuscript” and Referee 3 who states that “The data presented here are based on very challenging yet technically excellent experiments”.

- *“The manuscript is poorly written, unfinished or copied sentences, incorrect use of English language, not well-organized”*. We apologize for our incorrect use of English language. If the referee could document his comments and be more specific, it would help improve the quality of our manuscript. However, we would like to point out that we never faced this type of comments for any of our papers published in Nature Cell Biology, Nature Communications, The Journal of Cell Biology...

References from the lab on spindle morphogenesis:

- Meiotic spindle assembly and chromosome segregation in oocytes.

Bennabi I, Terret ME, Verlhac MH.

J Cell Biol. 2016 Dec 5;215(5):611-619. Epub 2016 Nov 22. Review.

- Rebuilding MTOCs upon centriole loss during mouse oogenesis.

Luksza M, Queguigner I, Verlhac MH, Brunet S.

Dev Biol. 2013 Oct 1;382(1):48-56. doi: 10.1016/j.ydbio.2013.07.029. Epub 2013 Aug 14.

- Using FRET to study RanGTP gradients in live mouse oocytes.

Dumont J, Verlhac MH.

Methods Mol Biol. 2013;957:107-20. doi: 10.1007/978-1-62703-191-2_7.

- Error-prone mammalian female meiosis from silencing the spindle assembly checkpoint without normal interkinetochore tension.

Kolano A, Brunet S, Silk AD, Cleveland DW, Verlhac MH.

Proc Natl Acad Sci U S A. 2012 Jul 3;109(27):E1858-67. doi: 10.1073/pnas.1204686109. Epub 2012 May 2.

- HURP permits MTOC sorting for robust meiotic spindle bipolarity, similar to extra centrosome clustering in cancer cells.

Breuer M, Kolano A, Kwon M, Li CC, Tsai TF, Pellman D, Brunet S, Verlhac MH.

J Cell Biol. 2010 Dec 27;191(7):1251-60. doi: 10.1083/jcb.201005065. Epub 2010 Dec 20.

- Meiotic regulation of TPX2 protein levels governs cell cycle progression in mouse oocytes.

Brunet S, Dumont J, Lee KW, Kinoshita K, Hikal P, Gruss OJ, Maro B, Verlhac MH.

PLoS One. 2008 Oct 3;3(10):e3338. doi: 10.1371/journal.pone.0003338.

- Interactions between chromosomes, microfilaments and microtubules revealed by the study of small GTPases in a big cell, the vertebrate oocyte.

Verlhac MH, Dumont J.

Mol Cell Endocrinol. 2008 Jan 30;282(1-2):12-7. doi: 10.1016/j.mce.2007.11.018. Epub 2007 Nov 22. Review.

- A centriole- and RanGTP-independent spindle assembly pathway in meiosis I of vertebrate oocytes.

Dumont J, Petri S, Pellegrin F, Terret ME, Bohnsack MT, Rassinier P, Georget V, Kalab P, Gruss OJ, Verlhac MH.

J Cell Biol. 2007 Jan 29;176(3):295-305.

- DOC1R: a MAP kinase substrate that control microtubule organization of metaphase II mouse oocytes.

Terret ME, Lefebvre C, Djiane A, Rassinier P, Moreau J, Maro B, Verlhac MH.

Development. 2003 Nov;130(21):5169-77. Epub 2003 Aug 27.

- Meiotic spindle stability depends on MAPK-interacting and spindle-stabilizing protein (MISS), a new MAPK substrate.

Lefebvre C, Terret ME, Djiane A, Rassinier P, Maro B, Verlhac MH.

J Cell Biol. 2002 May 13;157(4):603-13. Epub 2002 May 13.

Referee#2:

We thank the referee for his enthusiastic comments.

This study examines the effect of kinesin-14 overexpression and inhibition on meiotic spindle assembly and chromosome segregation in mouse oocytes. The main conclusion is that too much kinesin-14 dependent microtubule sliding in the early phase of meiotic spindle assembly causes spindles to bipolarize too quickly which leads to chromosome misalignment and segregation errors during anaphase. The study impresses by high quality experiments and careful and detailed quantitative analysis of the observations. It is a beautiful quantitative cell biology study. The results are novel, very interesting and significant. Overall the study is very well presented. An excellent manuscript.

This reviewer has only two major issues:

(1) Results: The authors perform a control experiment with a kinesin-14 mutant that is expected to crosslink microtubules but to lack the ability to slide microtubules. This is a nice experiment allowing a straight-forward conclusion about the importance of kinesin-14 motility for the observed effects, provided localization to the spindle is similar as for the wildtype. The authors state that this is the case, and cite Figures EV1B and 1E to support their claim. But it seems that Fig. 1E shows labelled microtubule fluorescence in the presence of the mutant (that is apparently not visualized) and Figure EV1B wildtype GFP-HSET fluorescence. A quantification as performed for the wildtype in Figure EV1D seems to be missing for the mutant, apparently leaving the claim unsupported.

We agree with the referee that we did not quantify HSET mutant localization to the spindle as it was done for HSET WT in Figure EV1D (on fixed samples, at NEBD+4h30, to compare it to endogenous HSET). However, we showed in live that GFP-HSET mutant is expressed on the spindle (Figure EV1E) as GFP-HSET WT (Figure EV1B, whose localization is similar to the endogenous), and even more overexpressed than GFP-HSET WT (Figure EV1F). Following the referee advice, we visualized in immunofluorescence HSET mutant, HSET WT and endogenous HSET at NEBD+4h30 in the same conditions and in parallel on fixed samples (new Figure EV1A). As shown in this new figure, HSET mutant localization is comparable to HSET WT and endogenous HSET localization, in agreement with our live experiments and to what was published before in HeLa cells (Cai, Mol Biol of the Cell 2009).

(2) Discussion: The most significant part of the Discussion seems to be the proposal that centrosomes were lost in oocytes to avoid fast mitosis-like spindle bipolarisation to allow for enough time for chromosome alignment. In a way that is a re-statement of the major result. What could turn this summary into a discussion would be a proposed explanation why mitosis-style bipolarization works for mitosis, but not for meiosis, i.e. what's the specific difference between meiosis and mitosis - probably related to chromosome properties - that requires such different mechanisms.

We thank the referee for his suggestion and changed the discussion accordingly.

Meiosis I is peculiar since homologous chromosomes linked by chiasmata progressively align on the metaphase plate, instead of single chromosomes in mitosis. Thus the volume, shape and occupancy of the objects (the chromosomes) moving towards the metaphase plate are completely different in meiosis, and chromosomes are active participants in meiotic spindle assembly (Radford 2017).

Minor issues:

(3) It is not always clear from the main text and the figure legends how experiments were performed, i.e. which molecules were labelled for microscopy, whether live or fixed (immunostained) specimen were observed and which type of microscopy was used (confocal, SIM?). It would be useful to have this information in each figure legend.

We agree with the referee that since we combine fixed and live samples for microscopy, it can be hard to follow. We changed all the figure legends where needed (Figures 1, EV1, 2, EV3, 3, 4, 5) as required by the referee.

(4) This reviewer found it odd that the data of Figure EV3B were presented in an accompanying figure, although they very directly show main results, whereas the results of the further analysis were presented in the main figure in a somewhat wordy manner. Can probably be streamlined.

We agree with the referee, but for space constraint, we cannot change that.

(5) Figures 3B and D seem to show more or less the same information - really necessary to show both?

We think that both figures are important, since they show two different things. The first one (Figure 3B) describes evolution of spindle length during meiosis I in HSET WT overexpressing oocytes (shown in Figure 3A). It highlights the fact that spindle length recovers in these oocytes despite the presence of extremely elongated spindles at the beginning of the process. The second one (Figure 3D) compares the initial and final spindle lengths of Control versus HSET WT overexpressing oocytes (shown in Figure 3C). It emphasizes the fact that the initial states are very different, but not the end states. For clarity and thanks to the referee suggestion, we modified the figure and removed the Ctrl from panel B.

(6) Methods: The description of the methods used for quantifications is rather brief. Some more detail or a link to the used scripts would be desirable.

We had a link for the scripts that did not appear in the previous version of the paper and we apologize for that. We added it back. The 3D_Spindle_Analysis plugin is available at: https://github.com/pmailly/3D_Spindle_Analysis

(7) Language: Abstract: 'extended and reversed engineered process' - unclear term; page 6: 'plasticity' - definition of what this means in this context is missing = unnecessary; some typos (e.g. Methods: 'Sutted filter sheel')

We fixed these errors and the manuscript was corrected by a native English speaker.

Referee#3:

We thank the referee for his positive and constructive comments.

Bennabi et al aim at analyzing further insights on meiotic spindle assembly in female oocytes. They use the established mouse oocyte system, which allows following the process in real time in a mammalian model. Several studies in recent years have highlighted the kinetic, conceptual and some molecular differences of mitotic vs meiotic spindle formation.

While centrosomes in somatic cells ensure rapid microtubule assembly and spindle pole formation, the absence of centrosomes in vertebrate oocytes necessitates a rather elongated "inside-out" spindle formation mechanism. In mouse oocytes, several non-centrosomal MTOCs are organised into a spindle pole by the action of microtubule-associated and motor proteins finally leading to spindle pole formation.

The authors come up with the intriguing idea to switch meiotic spindle formation into a more mitotic-like pathway using overexpression of the minus-end directed kinesin HSET/KifC5b. Indeed, increasing the levels of HSET leads to faster pole formation in mouse oocytes, somewhat reminiscent of rapid pole formation in mitotic cells. At the same time, the authors document that premature bipolarity gained by mild HSET overexpression compromises proper spindle function.

The data presented here are based on very challenging yet technically excellent experiments. I also have no doubt that this represents a potentially very interesting observation on meiotic spindle formation. However, the work does not unequivocally demonstrate the biological significance of the

role of HSET in shifting the system from meiotic to mitotic behaviour. Therefore, due to my opinion, the work is not suitable for publication in EMBO J.

My major concern is twofold:

1. The authors interpret accelerated pole formation after HSET overexpression as a shift towards "mitotic" behaviour. Such a shift would certainly be interesting as it could allow drawing conclusions about the driving force for the evolution of meiotic spindle formation. However, this is very difficult to recapitulate from the data presented here. I do concede that the HSET overexpression situation is well characterised but it is unclear to me if this situation really reflects a sort of mitotic pathway. Is there any evidence that levels/activity of HSET (e.g. compared to MT mass) are higher in somatic cells entering mitosis than in oocytes entering meiosis I? How is HSET regulated before meiosis and after fertilization?

Interestingly, some spindle assembly factors undergo a rise during meiosis I, such as TPX2 (Brunet, Plos One 2008; Chen, NCB 2013) or Miss (Lefebvre, J Cell Biol 2002). HSET follows the same pattern in oocytes (our manuscript, previous Figure 4A-B). Thus oocytes start meiosis I with low levels of key spindle assembly factors. On the contrary, HSET levels seem to be already high in somatic cells entering mitosis as HSET is sequestered in an active form in the nucleus during interphase and engages its microtubule targets upon nuclear envelope breakdown (Goshima, J Cell Biol 2005; Cai, Mol Biol Cell 2009; Hepperla, Dev Cell 2014). Raising the level of HSET early on in meiosis I would thus mimic a mitotic situation in terms of levels. To answer the referee's question, we performed additional experiments to monitor HSET levels in oocytes before meiosis (Prophase I) and after meiosis completion (after parthenogenetic activation mimicking fertilization). Levels of HSET are low in Prophase I (1.6 times lower than at NEBD+1h, see new Figure 4A-B). This confirms the fact that oocytes do enter meiotic divisions with low levels of HSET. Interestingly, HSET levels are 1.28 times higher after meiosis (activation of the oocytes mimicking fertilization) compared to Prophase I arrested oocytes (see new Figure 4A-B). In addition, HSET is strongly enriched in the pronucleus after meiosis (2.19 times more HSET in the pronucleus compared to the nucleus of Prophase I arrested oocytes, see new Figure 4A and quantification below).

Enrichment of endogenous HSET in the nucleus of Prophase I oocytes (before meiosis resumption) and in the female pronucleus of activated oocytes (after meiosis, mimicking fertilization), as quantified from immunofluorescent data.

Thus after fertilization of the oocyte by sperm, the zygote, comparable in size to the oocyte and also devoid of centrioles in rodents, enters the first mitotic division with more HSET than oocytes. Accordingly, spindle shape in the zygote is mitotic-like: elongated, with focused poles (Louvet-Vallée, Curr Biol 2005; Chaigne, Nat Commun 2016; see images below).

Live microscopy images of an oocyte in meiosis I and a zygote expressing GFP-EB3 (green, microtubules) and Histone-RFP (red, DNA). Scale bar 20 micrometers.

This suggests that HSET levels at M-phase entry matter for spindle shape, and we now discuss this point. We thank the referee for his suggestion.

2. Apart from the fact that its MT sliding activity is required for premature MTOC clustering, we do not really understand what increased HSET levels do at the molecular level / MT organization level and why the rather mild increase dramatically changes spindle formation. Modeling may be required to understand what is going on. Is it possible that other kinesin-14 family members may act similarly?

We agree with the referee that it is not clear why the rather mild increase in HSET levels dramatically changes spindle assembly. To answer this question, and as suggested by the referee, we have been collaborating for more than a year with François Nédélec and Serge Dmitrieff (EMBL) to model spindle formation in mouse oocytes. We are trying to recapitulate the whole process of spindle formation. It is an ongoing and demanding project that is unfortunately not finished yet and goes beyond this project.

Further concerns:

The HSET overexpression ratios are defined to be 1.6 while 4.2 fold more localizes to the spindle. How are the levels regulated and is there an explanation for the overrepresentation at the spindle? Is it just the mass of MT forming, or is there evidence for an additional level of regulation?

HSET is a microtubule binding protein, so its enrichment on the spindle is expected when slightly overexpressed. In addition, we have evidence that HSET overexpression does neither change microtubule dynamics (our FRAP experiments Figure 2F) nor the microtubule mass (see below a quantification of fluorescence intensity of EB3-GFP at NEBD in controls and HSET WT expressing oocytes).

Microtubule (MT) normalized signal intensity in HSET OE oocytes and controls expressing mCherry-EB3 was assessed in the chromosome vicinity at NEBD (inside the red circle on the scheme). Microtubules density is not significantly different in HSET OE oocytes compared to controls at NEBD. Not significant (n.s) P-value=0.35.

Alternatively, the progressive rise of Ran targets such as TPX2 (Brunet, Plos One 2008) during meiosis I could contribute to the spatial regulation of HSET within the spindle by the Ran-GTP gradient itself as shown in *Xenopus* egg extracts (Weaver, Curr Biol 2015). To test this hypothesis, we could modulate the levels of Ran-GTP as done before in the lab (Dumont, JCB 2007). However this also impacts microtubule density, and thus potentially HSET recruitment and would therefore be difficult to interpret. In addition, it was shown recently that other levels of regulation target proteins including HSET to the spindle in *Drosophila* oocytes (Beaven, JCB 2017).

Figure/graph EV2C does not display errors bars although statistical data is presented.

Graph EV2C does not display error bars because it represents percentages. Statistics on percentages can only be performed in experiments where the size of the sample (number of oocytes analysed) is identical in all conditions. This is clearly not the case for such difficult experiments using oocytes, with the paucity and variability in oocyte number.

Figure 4 shows endogenous HSET behaviour during MI "accumulating" 1.7 fold with time. I guess this refers to the overall HSET levels? It seems that spindle localisation is stronger. Could this be quantified as well? How is the increased binding interpreted?

The 1.7 fold accumulation with time indeed refers to the overall HSET levels. However, we quantified spindle localization, and it follows the same trend: HSET is accumulating 1.7 fold with time on the spindle (see below).

Enrichment of endogenous HSET on the spindle during meiosis I, as quantified from immunofluorescence data.

This increased binding can be a reflection of the increase in microtubule assembly throughout meiosis I (Brunet, Plos One 2008), without excluding spatial regulation of HSET within the spindle by the Ran-GTP gradient itself (Weaver, Curr Biol 2015). We thank the referee for his suggestion. Given the importance of the results on chromosome segregation, these results should be more clearly quantified.

Fig. 5C: do we see 18 vs. 26 spindles and respective alignment defects (0 in control, 9 or 10 after HSET wt injection?). Wouldn't this result be the quantification matching the defect shown in Fig. 5A?

Is it correct: the quantification means 15/16 of 26 spindles organized their chromosomes well in metaphase; do they still show chromosome segregation errors, like shown in Fig. 5B?

Figure 5C is indeed the quantification of extracted from Figure 5A. Oocytes that align their chromosomes properly in metaphase do not display chromosome segregation errors later in anaphase.

Several spelling and grammar errors need to be corrected.

We hopefully fixed these errors, as the manuscript was corrected by a native English speaker.

2nd Editorial Decision

11 December 2017

Thank you for the submission of your revised manuscript to our editorial offices. We have now received the reports from the two referees that were asked to re-evaluate your study (you will find enclosed below). These are the original referees #2 and #3 from your submission to The EMBO Journal. As you will see, both referees support the publication of your manuscript in EMBO reports. Referee #2 has some further suggestions to improve the paper that we ask you to address in a final revised version of the manuscript.

Further, I have these few editorial requests:

The title is currently too complicated and not fully comprehensible. Could you suggest and

alternative (without using more than 100 characters including spaces)?

Please correct the mistake in figure 4 F-G and the associated legend you pointed out to me earlier.

I look forward to seeing the final revised version of your manuscript when it is ready. Please let me know if you have questions or comments regarding the revision.

REFEREE REPORTS

Referee #1:

The authors have nicely addressed all my concerns (that were of a rather technical nature). This reviewer continues to think that this is a very original study providing interesting results supported by high quality data.

Referee #2:

Bennabi et al. hand in a revised version of their manuscript "Engineering mitotic-like mode of spindle assembly in oocytes leads to chromosome alignment defects" to EMBO Reports.

The new version addresses key issues that I had raised in my previous review. First of all, it provides evidence for rising levels of HSET during meiosis, i.e. after meiosis I. Likewise, I appreciate the data on increasing accumulation of HSET on the spindle. These observations also find reflection in the discussion. Together, this adds up to a, now more convincing, set of experiments that support the intriguing idea of HSET being a key component in changing the kinetics of spindle formation.

Second, the way writing the manuscript was considerably improved both with respect to correct English language but also in somewhat toning down the strict interpretation of a switch from meiotic to mitotic spindle assembly mechanisms. To further underline this last point, I make some suggestions for a few changes in wording that may still be considered prior to publication of the manuscript.

Introduction:

I suggest leaving out the phrase „thus forcing meiotic spindle morphogenesis to be more mitotic-like". To my opinion, the first half of the sentence "... accelerates spindle formation, in particular spindle bipolarisation and aMTOC clustering." is perfectly sufficient to describe what happens without anticipating a complicated and difficult interpretations.

Results:

The following change may be considered: "This suggests that, for the most part, changes in the timing of spindle bipolarization require microtubule sliding by HSET".

Likewise, the subheading "Shifting meiotic spindle assembly towards a mitotic mode" may be toned down, e.g. by saying "Shifting meiotic spindle morphology towards mitosis-like morphology". This is, in fact, what the authors use in the following lines and mostly throughout the manuscript to describe HSET-induced changes in spindle morphology.

I suggest using "consistently" instead of "accordingly" in the first line of p.8

Discussion... may also include a quantitative argument about HSET levels: mild overexpression may be comparable to the physiological transition from meiosis I to II and beyond, i.e. mitosis 1.

Following your requests, we have:

- changed the title.
- corrected the mistake in figure 4 F-G and its associated legend.
- incorporated all the suggestions of Referee #2.

Corresponding Author Name: Marie-Hélène VERLHAC and Marie-Emilie TERRET

Manuscript Number: EMBOR-2017-45225-T